# ZSQ4MIL: Zero-shot Quantization for Multiple Instance Learning

## Abstract

### Abstract

Zero-shot quantization (ZSQ) has emerged as a promising model compression paradigm that bypasses data privacy and security barriers in model deployment by leveraging synthesized samples for quantization calibration. Despite its remarkable success in various mainstream vision tasks, such as classification, detection, and segmentation, existing ZSQ approaches cannot be directly transferred to Multiple Instance Learning (MIL) models due to the unique hierarchical bag-instance structure inherent to MIL. To bridge this gap, this paper proposes ZSQ4MIL, the first ZSQ framework explicitly tailored for MIL. The core of ZSQ4MIL lies in synthesizing calibration data that perfectly aligns with the intrinsic distributional characteristics of MIL data. Specifically, we deeply analyze three core structural priors of instances within MIL bags: (i) instance heterogeneity, (ii) inter-class separability, and (iii) intra-class compactness. Empirical findings reveal that these structural priors fundamentally dictate the behavioral discrepancy of MIL models when processing random Gaussian noise versus real images. This insight inspires us to reconstruct MIL-aware calibration data by inverting optimized Gaussian noise. Methodologically, we first introduce an instance-level contrastive learning strategy to preserve feature heterogeneity. Subsequently, a grouped instance pseudo-labeling constraint is enforced to guarantee inter-class separability. Finally, a class-centric distance optimization scheme is proposed to further enhance intra-class compactness and widen inter-class margins. Based on these, we establish the first comprehensive quantization benchmarks for MIL. Extensive experiments thoroughly validate the effectiveness of ZSQ4MIL, which remarkably surpasses real-data-dependent calibration methods under several configurations. Code is available in the supplementary material.

## 1 Introduction

Multiple Instance Learning (MIL) has played a crucial role in numerous computer vision applications (Carbonneau et al., 2018; Gadermayr & Tschuchnig, 2024; Fatima et al., 2023; Yu et al., 2025; Jalil & Zafra, 2025). It adopts a bag-level annotation paradigm, where each bag is labeled while its constituent instances remain unlabeled. This formulation is particularly suited for scenarios where fine-grained annotations are costly or impractical to obtain. As a result, MIL has been widely applied to weakly supervised image retrieval and high-resolution image classification (Ling et al., 2024; Ilse et al., 2018), significantly reducing annotation efforts.

The increasing scale of neural networks and the growing demand for deploying deep models on resource-constrained edge devices have driven the development of model compression. Among model compression techniques (Chen et al., 2025a;b; Li et al., 2025b; Sun et al., 2024; Sreenivas et al., 2024), quantization is a key technique that reduces model size and computational cost while preserving the model architecture and its accuracy. However, conventional quantization typically relies on accessing the original training data,

which is often infeasible due to privacy and security concerns, or the overwhelming overhead associated with storing and transmitting large-scale datasets.

Zero-Shot Quantization (ZSQ) (Cai et al., 2020; Chen et al., 2024; Li et al., 2023; Shoukai et al., 2020) eliminates the need for real calibration data by generating synthetic samples to optimize quantization parameters. It has demonstrated great potential across standard tasks such as image classification, object detection, and semantic segmentation. Despite these successes, existing ZSQ paradigms cannot be directly extended to MIL.

For instance, methods like PSAQ (Li et al., 2023) primarily mine patch-level dependencies within a single isolated image. Conversely, approaches like IntraQ (Zhong et al., 2022) synthesize calibration data by matching general inter-image marginal distributions. Fundamentally, these conventional methods operate under an implicit independent and identically distributed (i.i.d.) assumption, treating individual images as discrete samples. Consequently, their data synthesis strategies are inherently misaligned with the unique hierarchical bag-instance structure of MIL. This underlying mismatch renders existing ZSQ methods incapable of capturing the specific intra-bag heterogeneity and inter-class separability dictated by weak bag-level annotations.

We address this gap by introducing ZSQ4MIL, a novel ZSQ framework that tackles MIL model quantization from the perspective of MIL-aware data reconstruction. *We begin by analyzing the key structural priors of instances within MIL bags, identifying three fundamental properties: (i) instance heterogeneity, (ii) inter-class separability, and (iii) intra-class compactness.* Through empirical studies, we demonstrate that these properties dictate the behavioral disparity of MIL models when processing random Gaussian noise versus real images. This mechanistic insight logically motivates our strategy to synthesize realistic MIL calibration data by progressively optimizing Gaussian noise.

To reconstruct data that strictly satisfies these structural priors, we propose a three-stage optimization pipeline. First, to preserve instance heterogeneity, we introduce an instance-level contrastive learning strategy that prevents mode collapse and encourages diverse feature patterns. Second, to enforce inter-class separability, a grouped pseudo-label constraint is designed to generate instance clusters that strictly align with the decision boundaries of the frozen MIL classifier. Finally, to enhance intra-class compactness and expand inter-class margins, we incorporate a class-centric distance optimization mechanism. Jointly, these components facilitate the faithful reconstruction of MIL-aware synthetic data, enabling highly accurate quantization parameter estimation.

To rigorously evaluate our approach, we establish comprehensive quantization benchmarks for MIL using multiple mainstream datasets and backbones. Extensive experiments under diverse quantization settings demonstrate the effectiveness of ZSQ4MIL. Notably, our method even matches or slightly surpasses real-data-dependent calibration in several challenging configurations. For example, when quantizing AB-MIL to W8A8, ZSQ4MIL yields over 30% absolute improvement compared to existing state-of-the-art ZSQ methods like IntraQ, completely closing the performance gap with real-data calibration.

Our main contributions are summarized as follows:

- We present the first zero-shot quantization framework specifically tailored for MIL. By identifying the statistical mismatch between noise and real MIL data, we formulate a synthesis strategy fundamentally grounded in MIL structural priors.

- We establish comprehensive, brand-new benchmarks for MIL quantization across multiple open-source datasets and representative MIL architectures, paving the way for future research in this domain.

- ZSQ4MIL achieves state-of-the-art zero-shot quantization performance, demonstrating robustness across various bit-widths and quantizers, and performing on par with real-data calibration methods.

# 2 Related Works

## 2.1 Data-Driven Quantization

Model quantization is conventionally dominated by data-driven paradigms, broadly bifurcated into Quantization-Aware Training (QAT) (Gong et al., 2019; Yin et al., 2019; Fan et al., 2020; Esser et al., 2019) and Post-Training Quantization (PTQ) (Nagel et al., 2020; Li et al., 2021; Wei et al., 2022). Both trajectories strictly depend on real calibration data to derive optimal parameters for mapping floating-point values to low-precision integers. QAT jointly optimizes model weights and quantization parameters across the full training set, often achieving extremely high precision at the cost of substantial computational overhead. Early QAT methods introduced the straight-through estimator (STE) (Yin et al., 2019) to bypass the non-differentiable rounding operation. To further minimize quantization error, LSQ (Esser et al., 2019) and LSQ+ (Bhalgat et al., 2020) incorporated step sizes and offsets as learnable parameters, while QuantNoise (Fan et al., 2020) leveraged random subset quantization to ensure unbiased gradient flows.

Conversely, PTQ avoids expensive retraining by calibrating quantization parameters using merely a small fraction of the training data. For instance, AdaRound (Nagel et al., 2020) demonstrated that simple nearest-rounding is suboptimal and elegantly reformulated it as a quadratic unconstrained binary optimization problem. BRECQ (Li et al., 2021) struck a balance between layer-wise and network-level optimization via block-wise reconstruction, whereas QDrop (Wei et al., 2022) introduced stochastic activation dropping to bolster quantization robustness.

Despite their impressive progress, the inevitable reliance of both PTQ and QAT on real data fundamentally restricts their applicability in data-restricted or privacy-sensitive deployment environments.

## 2.2 Zero-Shot Quantization

To circumvent data dependency, Zero-Shot Quantization (ZSQ) (Chen et al., 2023; 2024; Li et al., 2023; Shoukai et al., 2020) reconstructs synthetic calibration sets by exploiting the internal statistics of pre-trained models. Early milestones, such as ZeroQ (Cai et al., 2020) and GDFQ (Shoukai et al., 2020), predominantly leveraged Batch Normalization (BN) statistics embedded in CNNs, coupled with Inception-style losses, for data synthesis. To further enhance the diversity and robustness of synthetic samples, subsequent studies introduced adversarial learning strategies (ZAQ (Liu et al., 2021a)) or explicitly captured inter-image marginal distributions (IntraQ (Zhong et al., 2022)). As architectures evolved, ZSQ was progressively extended: PSAQ (Li et al., 2023) enabled data-free quantization for Vision Transformers via patch-similarity constraints; OuroMamba (Ramachandran et al., 2025) synthesized visual state space representations using contrastive learning; and Li et al. (2025a) broadened ZSQ to object detection through task-specific distillation.

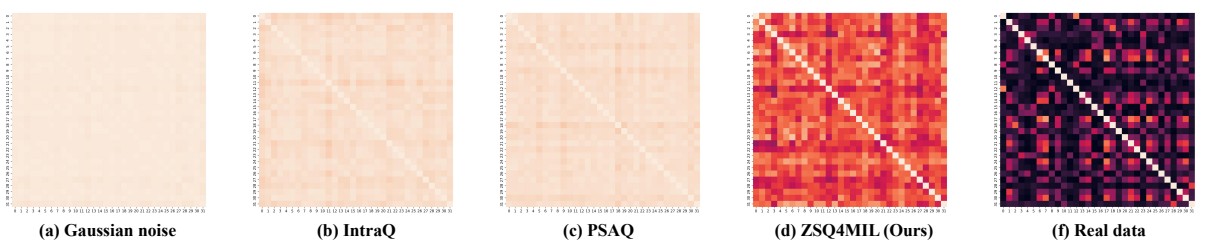

Figure 1: Cosine similarity matrices of instance features within a bag across different methods. As shown in (a)-(c), naive Gaussian noise and existing state-of-the-art ZSQ approaches (IntraQ and PSAQ) produce highly homogeneous instances, failing to capture intra-bag diversity. In contrast, our proposed ZSQ4MIL (d) effectively synthesizes heterogeneous instances, closely mimicking the rich feature diversity observed in the real data (f).

However, existing ZSQ data reconstruction paradigms are exclusively confined to instance-level or patch-level statistical alignment. They lack the mechanistic capacity to model the intrinsic, hierarchical bag-instance structural dependency that strictly defines MIL training and inference. To the best of our knowledge, no prior work has investigated zero-shot quantization for MIL models, leaving a significant research gap that our work aims to bridge.

## 3 Methods

### 3.1 Preliminaries

**Quantizer.** Quantization is a critical technique in reducing the computational footprint of deep neural networks by mapping floating-point parameters into fixed-point. In this work, the widely used asymmetric uniform quantizer is adopted. For any floating-point weight or activation tensor $x$, the quantization process is described by the following equations:

$$q = \text{round}\left(\frac{\text{clip}(x, l, u)}{s}\right),$$ (1)

where $\text{clip}(x, l, u) = \min(\max(x, l), u)$ clips the values of $x$ to the range $[l, u]$, and $s = \frac{u-l}{2^b-1}$ is the scaling factor that maps floating-point values to fixed-point integers, with $b$ being the bit-width. The dequantized value $\overline{x}$ could be computed as:

$$\overline{x} = q \cdot s.$$ (2)

**Multiple instance learning (MIL).** Multiple instance learning is a weakly supervised learning framework where the model is provided with bags of instances, but only the labels for the bags are known. The labels for individual instances within each bag are unobserved. The objective of the MIL task is to predict the label of each bag, based on the instances it contains.

Formally, let $\{\boldsymbol{X}_1, \boldsymbol{X}_2, \ldots, \boldsymbol{X}_N\}$ denote a collection of $N$ bags, where each bag $\boldsymbol{X}_b$ contains multiple instances $\{\boldsymbol{x}_{b,1}, \boldsymbol{x}_{b,2}, \ldots, \boldsymbol{x}_{b,N_b}\}$. The task is to predict the label $Y_b$ for each bag. For binary classification, the label for the bag is defined as:

$$Y_b = \max_i\{y_{b,i}\},$$ (3)

where $y_{b,i}$ is the label of the $i$-th instance in the $b$-th bag. Specifically, if any instance in the bag is positive ($y_{b,i} = 1$), then the entire bag is considered positive ($Y_b = 1$). If all instances in the bag are negative ($y_{b,i} = 0$ for all $i$), then the bag label is negative ($Y_b = 0$).

MIL is widely used in tasks where instance-level annotations are expensive or infeasible, such as image set classification or whole-slide image analysis. A typical MIL model consists of three key components: an encoder that extracts instance-level features, an aggregator (e.g., mean pooling, attention, or transformer-based) that combines instance representations into a bag-level embedding, and a bag classifier that outputs the final bag label.

### 3.2 Instance-Level Contrastive Learning for Heterogeneity

**Observation 1: heterogeneity of instances.** Our empirical analysis in Figure 1(a) reveals a critical deficiency in naive Gaussian noise for quantization calibration: its instances are statistically homogeneous. When processed by a pretrained MIL feature extractor, Gaussian noise yields highly similar embeddings, failing to emulate the rich diversity of real-world instances. In contrast, real instances present profound heterogeneity, as demonstrated in Figure 1(f). Motivated by this, we synthesize a set of instances that are individually distinct and collectively diverse in the feature space.

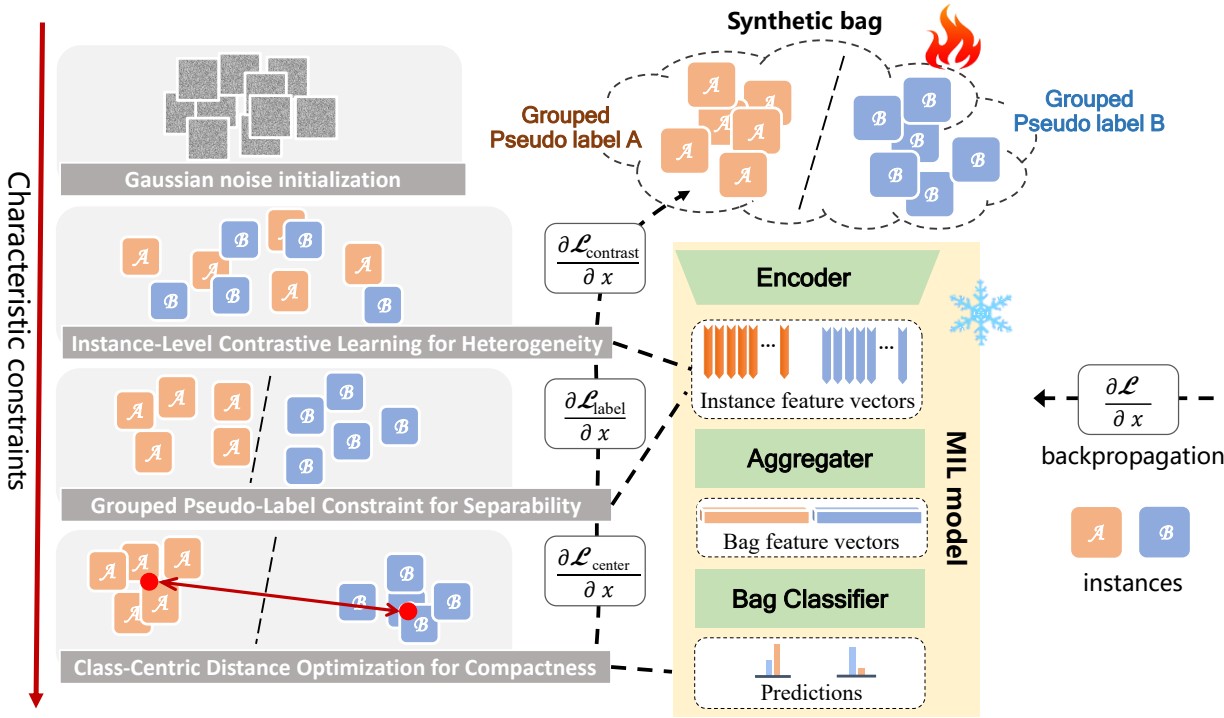

Figure 2: An overview of the proposed framework ZSQ4MIL.

**Formulation.** To this end, we introduce an instance-level contrastive learning strategy. This strategy enforces that different augmented views of the same synthesized instance produce similar representations, while being dissimilar to views from other instances. Let $\boldsymbol{X} = \{\boldsymbol{x}_i\}_{i=1}^N$ be the batch of $N$ synthesizable instances, initialized from Gaussian noise. We define a stochastic augmentation module, $\mathcal{T}$, which generates two distinct correlated views for each instance: a global view $\boldsymbol{x}_i^g = \mathcal{T}_g(\boldsymbol{x}_i)$ and a local view $\boldsymbol{x}_i^l = \mathcal{T}_l(\boldsymbol{x}_i)$.

Let $f_\theta(\cdot)$ denote the frozen feature extractor of the MIL model. We apply the contrastive loss directly to the output features. The features are thus given by $\boldsymbol{f}_i^g = f_\theta(\boldsymbol{x}_i^g)$ and $\boldsymbol{f}_i^l = f_\theta(\boldsymbol{x}_i^l)$. We then employ the InfoNCE loss to maximize the agreement between positive pairs $(\boldsymbol{f}_i^l, \boldsymbol{f}_i^g)$ against all other negative pairs in the batch. The loss is formulated as:

$$\mathcal{L}_{\text{contrast}} = -\sum_{i=1}^N \log \frac{\exp(\text{sim}(\boldsymbol{f}_i^l, \boldsymbol{f}_i^g)/\tau)}{\sum_{j=1}^N \exp(\text{sim}(\boldsymbol{f}_i^l, \boldsymbol{f}_j^g)/\tau)}, \tag{4}$$

where $\text{sim}(\boldsymbol{u}, \boldsymbol{v}) = \boldsymbol{u}^\top \boldsymbol{v}$ denotes the dot product, and $\tau$ is a temperature hyperparameter. By backpropagating the gradients of $\mathcal{L}_{\text{contrast}}$ to the input instances $\boldsymbol{X}$, we force them to evolve from simple noise into a set of semantically heterogeneous patterns.

### 3.3 Grouped Pseudo-Label Constraint for Inter-class Separability

**Observation 2: inter-class separability of instances.** Although instance-level labels are inherently unobservable during MIL, empirical evidence suggests that the latent features of real instances remain highly separable. A faithful calibration set for quantization must, therefore, encapsulate both positive and negative instance representations to ensure accurate bag-level predictions.

**Formulation.** Thanks to the bag-level prediction head of the MIL model, we can ensure that our synthetic instances contain positive and negative ones. We achieve this by imposing a pseudo-label constraint at the bag level. The batch of $N$ synthetic instances $\boldsymbol{X}$ is partitioned into two disjoint subsets of equal size, which we designate as a pseudo-positive bag $\mathcal{B}_{\text{pos}}$ and a pseudo-negative bag $\mathcal{B}_{\text{neg}}$. We then feed these

synthetic bags into the complete frozen MIL model $M_\theta(\cdot)$, which outputs a bag-level probability. Our goal is to optimize the instances such that $M_\theta(\mathcal{B}_{\text{pos}})$ predicts a positive label (probability approaching 1) and $M_\theta(\mathcal{B}_{\text{neg}})$ predicts a negative label (probability approaching 0). This objective is enforced using a standard Binary Cross-Entropy (BCE) loss:

$$\mathcal{L}_{\text{label}} = \text{BCE}(M_\theta(\mathcal{B}_{\text{pos}}), 1) + \text{BCE}(M_\theta(\mathcal{B}_{\text{neg}}), 0). \tag{5}$$

This loss directly steers the synthesis process to generate instances that align with the decision boundaries learned by the MIL model, thereby ensuring inter-class separability at the model's output layer.

### 3.4 Class-Centric Distance Optimization for Intra-class Compactness

**Observation 3: intra-class compactness of instances.** While the grouped pseudo-label constraint promotes basic instance separability, real-world instances are structurally characterized by large inter-class margins and tightly compact intra-class distributions. Accurate quantization necessitates that this specific topological structure be explicitly captured by the calibration set.

**Formulation.** To better approximate the organization of real data and reinforce intra-class compactness within the calibration set, we explicitly impose a margin constraint between the feature clusters corresponding to pseudo-positive and pseudo-negative bags. We introduce a class-centric distance optimization scheme that operates directly on the instance features extracted by $f_\theta(\cdot)$. For the pseudo-positive and pseudo-negative bags, we compute their respective feature centroids:

$$\boldsymbol{c}_{\text{pos}} = \frac{1}{|\mathcal{B}_{\text{pos}}|} \sum_{\boldsymbol{x}_i \in \mathcal{B}_{\text{pos}}} f_\theta(\boldsymbol{x}_i), \quad \boldsymbol{c}_{\text{neg}} = \frac{1}{|\mathcal{B}_{\text{neg}}|} \sum_{\boldsymbol{x}_i \in \mathcal{B}_{\text{neg}}} f_\theta(\boldsymbol{x}_i). \tag{6}$$

To maximize the separation between these two conceptual classes, we employ a margin-based loss that pushes their centroids apart. The loss is defined as:

$$\mathcal{L}_{\text{center}} = \max(0, m - \|\boldsymbol{c}_{\text{pos}} - \boldsymbol{c}_{\text{neg}}\|_2), \tag{7}$$

where $m$ is a predefined margin for separation. This loss penalizes the configuration only if the distance between the centroids is smaller than $m$, effectively creating a large-margin separation in the feature space. This not only enhances inter-class separability but also promotes intra-class compactness implicitly by pulling instances towards their assigned cluster center.

### 3.5 The Overall Pipeline

**Total objective.** The overall objective function is a combination of the instance heterogeneity loss, the pseudo-label separability loss, the class-centric distance loss, and a total variation (TV) regularizer $\mathcal{L}_{\text{TV}}$ that encourages spatial smoothness in the generated images. The final objective is:

$$\mathcal{L}_{\text{total}} = \mathcal{L}_{\text{label}} + \mathcal{L}_{\text{contrast}} + \mathcal{L}_{\text{center}} + \mathcal{L}_{\text{TV}}. \tag{8}$$

**Synthetic data generation.** Our MIL-aware image synthesis framework integrates the three aforementioned objectives into a unified optimization process. The procedure begins by initializing a tensor $\boldsymbol{X}^{(0)} \in \mathbb{R}^{N \times C \times H \times W}$ with random Gaussian noise. This tensor is the sole trainable parameter in the synthesis phase; the MIL model $M_\theta$ remains frozen. At each iteration $t$, we update the instance tensor $\boldsymbol{X}^{(t)}$ by minimizing a comprehensive loss function $\mathcal{L}_{\text{total}}$ that combines our proposed constraints.

The gradient of $\mathcal{L}_{\text{total}}$ is computed with respect to the input tensor $\boldsymbol{X}$ and used to update it via an Adam optimizer. After a fixed number of iterations, the optimized tensor $\boldsymbol{X}^*$ serves as the synthetic calibration data.

**Model quantization.** With the generated data $\boldsymbol{X}^*$ in hand, we proceed with standard post-training quantization. The synthetic instances are formed into bags and fed through the full-precision model to collect activation statistics. These statistics are then used to determine the quantization parameters for both weights and activations, enabling the deployment of an efficient, quantized MIL model without ever accessing real data.

## 4 Experiments

To rigorously validate the efficacy of our proposed ZSQ4MIL framework, we designed a series of comprehensive experiments. This section first introduces the evaluation setup, including the construction of benchmark datasets, baseline methods, and implementation details. Subsequently, we present extensive performance comparisons across multiple dimensions and conduct an in-depth ablation study to isolate the contribution of each key algorithmic component.

### 4.1 Datasets and Benchmarks

We construct three MIL binary classification benchmarks based on widely-used public datasets: **MNIST**, **Bird vs Drone**, and **Cat vs Dog**. To simulate realistic MIL application scenarios, we carefully configure the dataset generation protocols. Specifically, the length of each bag (i.e., the number of instances) is sampled from a normal distribution with a mean of 100 and a standard deviation of 10, clamped within a range of 5 to 25,000. We generate 500, 100, and 1,000 bags for the training, validation, and testing sets, respectively. In each positive bag, the maximum number of target instances is set to 10. The bag label follows the standard MIL assumption: a bag is positive if at least one instance within it belongs to the target class; otherwise, the bag is negative. This setup not only establishes a standardized evaluation platform for future research but also introduces the "maximum number of target instances per bag" as a controllable variable, enabling us to systematically investigate algorithmic robustness under varying degrees of task sparsity.

### 4.2 Baselines

To benchmark the performance of ZSQ4MIL, we compare it against five representative calibration data settings:

- **Real data**: Utilizes real calibration data, representing the performance upper bound for post-training quantization.

- **Gaussian noise**: Employs unoptimized, raw Gaussian noise as calibration data, serving as a rudimentary zero-shot baseline.

- **Zero-padding**: Uses zero matrices devoid of any semantic or statistical information, acting as an extremely weak baseline.

- **PSAQ (Li et al., 2023)**: A patch-similarity-aware data-free quantization method that captures intra-image characteristics.

- **IntraQ (Zhong et al., 2022)**: A state-of-the-art ZSQ method that synthesizes calibration data by matching inter-image marginal distributions.

We conduct evaluations on several mainstream MIL backbones, including Mean-MIL, Max-MIL, and AB-MIL (Ilse et al., 2018), equipped with Swin-T (Liu et al., 2021b) and DeiT-B (Touvron et al., 2021) encoders pre-trained on ImageNet. All experiments report the highest accuracy and F1-score achieved on the test set.

### 4.3 Implementation Details

All experiments are implemented using the PyTorch framework and conducted on a single NVIDIA A6000 GPU. The baseline MIL models are trained with the Adam optimizer, utilizing a learning rate of $5 \times 10^{-4}$

and a momentum of 0.9 over 10 epochs. During the synthetic data generation phase, the Adam optimizer is also employed to iteratively optimize the Gaussian noise over 10,000 iterations with a batch size of 32. Unless otherwise specified, the maximum number of positive instances per bag is fixed at 10. We perform quantization using the quantizer detailed in the preliminary section. The notation $W(BW_1)A(BW_2)$ denotes that the model weights are quantized to $BW_1$ bits and the activations to $BW_2$ bits.

### 4.4 Performance Comparison

**Results on representative MIL architectures.** As shown in Table 1, ZSQ4MIL consistently mitigates the severe accuracy degradation induced by the absence of real calibration data across all representative MIL architectures. Notably, AB-MIL, which features an intricate attention-based aggregation mechanism, is particularly sensitive to calibration data quality. Under this challenging setup, conventional zero-shot baselines (Gaussian noise and zero padding) collapse entirely. Furthermore, advanced ZSQ methods like PSAQ and IntraQ also struggle significantly, yielding F1-scores below 46.00% on AB-MIL. This catastrophic failure occurs because attention aggregators rely heavily on the relative distribution and semantic contrast of features within a bag; synthesizing images as isolated instances fails to activate these aggregators properly. In stark contrast, ZSQ4MIL successfully reconstructs this intra-bag geometry, restoring performance closely to its full-precision counterpart with a 94.69% F1-score at W8A8 and maintaining 93.12% at W4A8. Interestingly, our method occasionally outperforms real-data calibration (e.g., 93.12% vs. 92.70% on AB-MIL with W4A8). This remarkable phenomenon suggests that the MIL-aware synthesized bags act as a data-agnostic regularizer. Instead of inheriting the long-tail biases or ambiguous boundary samples endemic

Table 1: Quantization results across representative MIL architectures on the Cats vs Dogs dataset. The MIL model utilizes the Swin-T backbone. "Acc." denotes Accuracy (%), and "F1" represents F1-score (%).

| Model | Methods | Zero shot | Precision | Acc. | F1 |
|---|---|---|---|---|---|
| | **Full precision** | − | **FP32** | **93.60** | **93.75** |
| AB-MIL | Real data | × | W8/A8 | 93.60 | 93.75 |
| | Gaussian noise | ✓ | W8/A8 | 50.00 | 0.00 |
| | Zero padding | ✓ | W8/A8 | 50.00 | 66.67 |
| | PSAQ | ✓ | W8/A8 | 63.20 | 41.77 |
| | IntraQ | ✓ | W8/A8 | 60.50 | 34.71 |
| | **ZSQ4MIL (Ours)** | ✓ | **W8/A8** | **94.50** | **94.69** |
| | Real data | × | W4/A8 | 92.60 | 92.70 |
| | Gaussian noise | ✓ | W4/A8 | 50.00 | 0.00 |
| | Zero padding | ✓ | W4/A8 | 50.00 | 66.67 |
| | PSAQ | ✓ | W4/A8 | 64.90 | 45.92 |
| | IntraQ | ✓ | W4/A8 | 62.90 | 41.02 |
| | **ZSQ4MIL (Ours)** | ✓ | **W4/A8** | **93.00** | **93.12** |
| | **Full precision** | − | **FP32** | **88.80** | **87.44** |
| Mean-MIL | Real data | × | W8/A8 | 86.70 | 87.99 |
| | Gaussian noise | ✓ | W8/A8 | 50.00 | 66.67 |
| | Zero padding | ✓ | W8/A8 | 50.00 | 66.67 |
| | PSAQ | ✓ | W8/A8 | 76.30 | 69.02 |
| | IntraQ | ✓ | W8/A8 | 82.10 | 78.30 |
| | **ZSQ4MIL (Ours)** | ✓ | **W8/A8** | **88.30** | **86.84** |
| | Real data | × | W4/A8 | 90.20 | 90.65 |
| | Gaussian noise | ✓ | W4/A8 | 50.00 | 66.67 |
| | Zero padding | ✓ | W4/A8 | 50.00 | 66.67 |
| | PSAQ | ✓ | W4/A8 | 77.20 | 70.47 |
| | IntraQ | ✓ | W4/A8 | 78.70 | 72.94 |
| | **ZSQ4MIL (Ours)** | ✓ | **W4/A8** | **88.90** | **83.26** |
| | **Full precision** | − | **FP32** | **94.90** | **95.08** |
| Max-MIL | Real data | × | W8/A8 | 94.80 | 94.98 |
| | Gaussian noise | ✓ | W8/A8 | 71.20 | 62.40 |
| | Zero padding | ✓ | W8/A8 | 50.00 | 66.67 |
| | PSAQ | ✓ | W8/A8 | 93.10 | 92.59 |
| | IntraQ | ✓ | W8/A8 | 92.90 | 92.36 |
| | **ZSQ4MIL (Ours)** | ✓ | **W8/A8** | **94.00** | **93.62** |
| | Real data | × | W4/A8 | 94.40 | 94.57 |
| | Gaussian noise | ✓ | W4/A8 | 59.80 | 66.56 |
| | Zero padding | ✓ | W4/A8 | 50.00 | 66.67 |
| | PSAQ | ✓ | W4/A8 | 89.80 | 89.55 |
| | IntraQ | ✓ | W4/A8 | 92.60 | 92.24 |
| | **ZSQ4MIL (Ours)** | ✓ | **W4/A8** | **93.60** | **93.16** |

to limited real datasets, ZSQ4MIL generates a uniformly distributed feature coverage that strictly adheres to the model's decision boundaries, thereby mitigating overfitting during the quantization parameter search.

**Results on various datasets.** As summarized in Table 2, ZSQ4MIL exhibits consistent superiority across datasets of varying visual complexity, ranging from local texture recognition (MNIST4MIL) to semantically rich object discrimination (Birds vs Drones and Cats vs Dogs).

Under the W8A8 configuration, ZSQ4MIL achieves performance nearly identical to its real-data calibrated counterpart across all datasets, with an average F1-score gap of less than 1%. On the challenging Cats vs Dogs dataset, ZSQ4MIL attains an F1-score of 95.23%, closely matching the 95.55% obtained using real data. Conversely, state-of-the-art ZSQ methods like PSAQ and IntraQ suffer severe performance drops (e.g., near-zero F1-scores on MNIST4MIL). Because these methods are tailored for standard single-instance classification, they indiscriminately force individual images to match global dataset statistics, completely destroying the sparse, weakly-supervised topology of MIL bags. These results underscore the strong generalization capability of ZSQ4MIL across different domains.

**Results under different difficulty settings.** We modulate task difficulty by controlling the maximum number of positive instances per bag on the Cats vs Dogs dataset. In MIL, extreme sparsity (fewer target instances) is notoriously challenging. As summarized in Table 3, across the entire difficulty spectrum (max positives = 10, 20, 40, 60), naive zero-shot baselines fail catastrophically. While PSAQ and IntraQ show mild improvements when the bag composition becomes denser and easier (e.g., IntraQ reaches 85.71% F1 at max positives = 60), they still fall significantly short. ZSQ4MIL, however, exhibits remarkable robustness, maintaining high accuracy even under the strictest sparsity setting (max positives = 10).

Importantly, as the task simplifies, ZSQ4MIL tracks and sometimes marginally exceeds full real-data calibration (e.g., 94.98% vs. 94.37% at max positives = 60). Denser positive-instance configurations reduce the ambiguity of bag-level gradients during synthesis, allowing ZSQ4MIL to reconstruct highly precise discriminative feature statistics that perfectly align the quantized network with the full-precision model.

Table 2: Quantization results across various datasets of varying visual complexity. The experiments involve quantizing the AB-MIL model with a DeiT-B encoder to the W8A8 configuration to thoroughly evaluate cross-domain generalization capabilities.

| Dataset | Methods | Zero shot | Acc. | F1 |
|---|---|---|---|---|
| MNIST4MIL | Full precision | – | 96.10 | 95.95 |
| | Real data | × | 94.90 | 94.78 |
| | Gaussian noise | ✓ | 50.00 | 0.00 |
| | Zero padding | ✓ | 50.00 | 0.00 |
| | PSAQ | ✓ | 50.46 | 2.37 |
| | IntraQ | ✓ | 50.10 | 0.40 |
| | **ZSQ4MIL (Ours)** | ✓ | **95.10** | **94.86** |
| Birds vs Drones | Full precision | – | 94.00 | 93.62 |
| | Real data | × | 93.60 | 93.16 |
| | Gaussian noise | ✓ | 50.00 | 0.00 |
| | Zero padding | ✓ | 50.00 | 0.00 |
| | PSAQ | ✓ | 75.90 | 68.25 |
| | IntraQ | ✓ | 78.10 | 71.96 |
| | **ZSQ4MIL (Ours)** | ✓ | **91.70** | **90.95** |
| Cats vs Dogs | Full precision | – | 94.00 | 94.16 |
| | Real data | × | 95.50 | 95.55 |
| | Gaussian noise | ✓ | 50.00 | 0.00 |
| | Zero padding | ✓ | 50.00 | 66.67 |
| | PSAQ | ✓ | 86.00 | 83.72 |
| | IntraQ | ✓ | 82.30 | 78.49 |
| | **ZSQ4MIL (Ours)** | ✓ | **95.20** | **95.23** |

Table 3: Quantization results across different maximum numbers of target instances. The experiments involve quantizing the AB-MIL model with a Swin-T encoder to the W8A8 configuration on the Cats vs Dogs dataset.

| Max num. | Methods | Zero shot | Acc. | F1 |
|---|---|---|---|---|
| | Full precision | – | 93.60 | 93.75 |
| 10 | Real data | × | 93.60 | 93.75 |
| | Gaussian noise | ✓ | 50.00 | 0.00 |
| | Zero padding | ✓ | 50.00 | 66.67 |
| | PSAQ | ✓ | 63.20 | 41.77 |
| | IntraQ | ✓ | 60.50 | 34.71 |
| | **ZSQ4MIL (Ours)** | ✓ | **94.50** | **94.69** |
| | Full precision | – | 94.00 | 94.16 |
| 20 | Real data | × | 94.10 | 94.27 |
| | Gaussian noise | ✓ | 50.00 | 0.00 |
| | Zero padding | ✓ | 50.00 | 66.67 |
| | PSAQ | ✓ | 69.10 | 55.28 |
| | IntraQ | ✓ | 76.60 | 69.45 |
| | **ZSQ4MIL (Ours)** | ✓ | **94.50** | **94.68** |
| | Full precision | – | 94.10 | 94.27 |
| 40 | Real data | × | 94.20 | 94.37 |
| | Gaussian noise | ✓ | 50.00 | 0.00 |
| | Zero padding | ✓ | 50.00 | 66.67 |
| | PSAQ | ✓ | 78.60 | 72.77 |
| | IntraQ | ✓ | 80.90 | 76.39 |
| | **ZSQ4MIL (Ours)** | ✓ | **94.90** | **95.08** |
| | Full precision | – | 94.20 | 94.37 |
| 60 | Real data | × | 94.60 | 94.78 |
| | Gaussian noise | ✓ | 50.00 | 0.00 |
| | Zero padding | ✓ | 50.00 | 66.67 |
| | PSAQ | ✓ | 85.10 | 82.49 |
| | IntraQ | ✓ | 87.50 | 85.71 |
| | **ZSQ4MIL (Ours)** | ✓ | **94.80** | **94.98** |

**Results using different quantizers.** To evaluate the universality of our approach, we test ZSQ4MIL with several representative PTQ quantizers, including MinMax, SmoothQuant (Xiao et al., 2023), and AWQ (Lin et al., 2024), as summarized in Table 4.

Across all quantizers and MIL architectures, ZSQ4MIL consistently delivers quantized models that retain near full-precision performance. Because our framework operates entirely at the data synthesis level rather than modifying the underlying quantization algorithm, it provides a high-fidelity proxy for activation distributions, satisfying the calibration prerequisites of any PTQ strategy. On AB-MIL, all quantizers achieve F1-scores above 93%. These results unequivocally demonstrate that ZSQ4MIL is quantizer-agnostic and can be seamlessly deployed as a plug-and-play module.

Table 4: Quantization results across different quantizers. The experiments involve quantizing the MIL model with Swin-T encoder to W8A8 configuration on the Cats vs Dogs dataset.

| Model | Quantizer | Acc. | F1 |
|---|---|---|---|
| AB-MIL | **Full precision** | **93.60** | **93.75** |
| | Max-Min | 94.50 | 94.69 |
| | SmoothQuant | 94.30 | 94.47 |
| | AWQ | 93.10 | 93.23 |
| Mean-MIL | **Full precision** | **88.80** | **87.44** |
| | Max-Min | 88.30 | 86.84 |
| | SmoothQuant | 90.30 | 90.48 |
| | AWQ | 88.30 | 86.81 |
| Max-MIL | **Full precision** | **94.90** | **95.08** |
| | Max-Min | 94.00 | 93.62 |
| | SmoothQuant | 93.60 | 93.16 |
| | AWQ | 94.50 | 94.68 |

Table 5: Ablation study on components of ZSQ4MIL. The experiments involve quantizing the MIL model with Swin-T encoder to W8A8 configuration on the Cats vs Dogs dataset.

| Model | $\mathcal{L}_{\text{label}}$ | $\mathcal{L}_{\text{centor}}$ | $\mathcal{L}_{\text{contrast}}$ | Acc. | F1 |
|---|---|---|---|---|---|
| AB-MIL | ✓ | ✓ | ✓ | **94.50** | **94.69** |
| | ✓ | ✓ | × | 66.70 | 50.07 |
| | ✓ | × | ✓ | 92.30 | 90.47 |
| | × | ✓ | ✓ | 93.50 | 92.68 |
| Mean-MIL | ✓ | ✓ | ✓ | **88.30** | **86.84** |
| | ✓ | ✓ | × | 75.10 | 66.84 |
| | ✓ | × | ✓ | 80.50 | 75.84 |
| | × | ✓ | ✓ | 82.00 | 78.10 |
| Max-MIL | ✓ | ✓ | ✓ | **94.00** | **93.62** |
| | ✓ | ✓ | × | 74.60 | 65.95 |
| | ✓ | × | ✓ | 92.60 | 92.59 |
| | × | ✓ | ✓ | 92.40 | 92.50 |

### 4.5 Discussion

**Ablation study on components.** To strictly isolate the contribution of each core component in ZSQ4MIL, we conduct detailed ablation experiments (Table 5) by progressively removing the proposed losses from the synthesis pipeline.

- **Instance-level contrastive loss** ($\mathcal{L}_{\text{contrast}}$): Its removal leads to a catastrophic mode collapse during synthesis (e.g., the F1-score of AB-MIL plummets from 94.69% to 50.07%). Without it, the optimized instances become homogeneous, proving that intra-bag diversity is the fundamental prerequisite for correctly activating the MIL aggregator.

- **Grouped pseudo-label constraint** ($\mathcal{L}_{\text{label}}$): Promotes inter-class separability. Without it, the synthesized bag fails to produce instances that distinctly trigger the positive and negative decision boundaries of the classifier.

- **Class-centric distance optimization** ($\mathcal{L}_{\text{center}}$): Enforces larger feature-space margins. Removing it results in tightly packed, fragile feature clusters that easily cross decision boundaries during the mapping from floating-point to low-precision integers, thereby severely harming quantization robustness.

The full model consistently yields the best performance, confirming that these three constraints act synergistically: diversity provides the foundational structure, pseudo-labels guide the semantic direction, and distance optimization ensures precision robustness.

**Visualization of synthetic images.** As illustrated in Figure 3, our proposed ZSQ4MIL (top row) effectively synthesizes diverse instances that perceptibly capture semantic archetypes (e.g., cat and dog facial features) encoded by the frozen MIL model, successfully decoding discriminative visual patterns. Conversely,

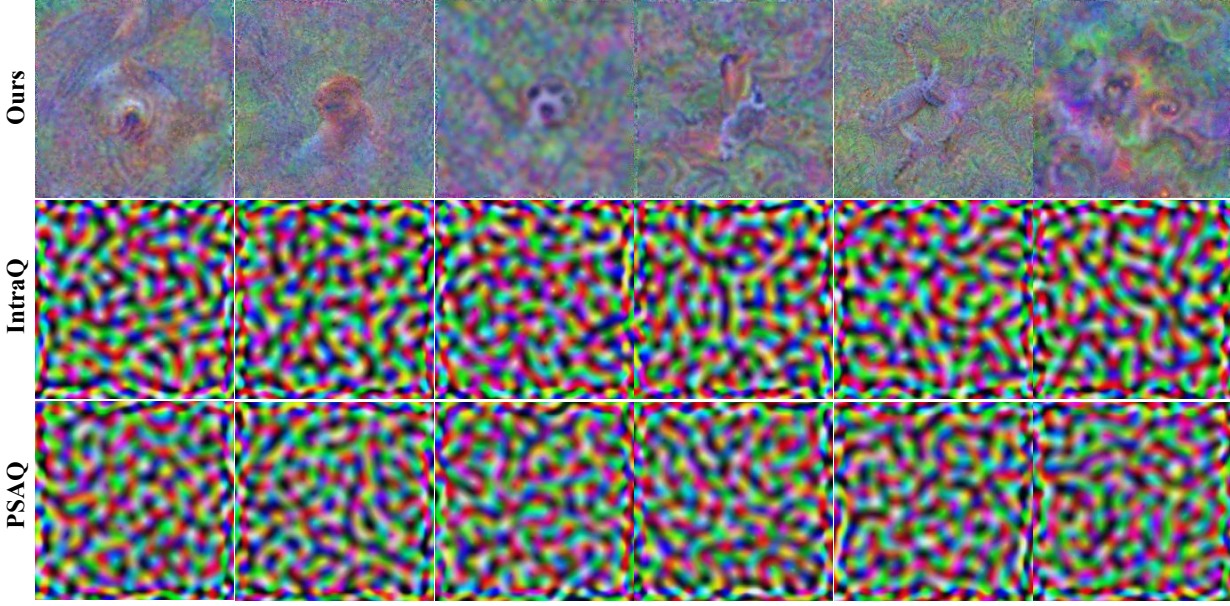

Figure 3: Visualization of synthetic instance bags generated by different zero-shot quantization methods (top: ZSQ4MIL, middle: IntraQ, bottom: PSAQ) on the Cats vs Dogs dataset.

existing ZSQ baselines such as IntraQ and PSAQ (middle and bottom rows) degenerate into meaningless, highly homogeneous, non-semantic high-contrast and low-contrast texture noise patterns. These baseline methods fail to extract meaningful semantic content from the frozen encoder, perfectly matching the feature-space homogeneity observed in Figure 1 and corroborating their inability to navigate the MIL bag-instance topology. Importantly, our synthesized images are decoupled from visual fidelity and remain visually distinct from real samples. This visual artifact is highly advantageous: it proves that our synthesis process is driven purely by the decision boundaries of the frozen network rather than reconstructing actual training data. Consequently, ZSQ4MIL inherently guarantees privacy preservation, extracting essential statistical properties for calibration without ever exposing sensitive, human-interpretable details.

## 5 Conclusion

This paper presents ZSQ4MIL, the pioneering zero-shot quantization framework explicitly tailored for multiple instance learning (MIL). By investigating the behavioral discrepancy of MIL models when processing real versus noise instances, we identify three fundamental structural priors inherent to MIL data: instance heterogeneity, inter-class separability, and intra-class compactness. Guided by these mechanistic insights, we formulate a novel data reconstruction pipeline—integrating instance-level contrastive learning, grouped pseudo-labeling, and class-centric distance optimization—to synthesize MIL-aware calibration bags entirely from Gaussian noise. While existing state-of-the-art ZSQ approaches are bottlenecked by an implicit i.i.d. assumption and consequently fail to capture the unique bag-instance topology, ZSQ4MIL overcomes this fundamental limitation by generating semantically coherent and structurally faithful calibration sets. Extensive experiments across diverse datasets, aggregation architectures, PTQ quantizers, and task sparsity settings unequivocally demonstrate that ZSQ4MIL consistently eclipses conventional zero-shot baselines and, remarkably, achieves parity with real-data-dependent calibration methods. Ultimately, this work establishes a brand-new benchmark and opens promising avenues for the privacy-preserving, resource-efficient deployment of MIL models in data-restricted edge environments.

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
