# OpenReview forum: "ZSQ4MIL: Zero-shot Quantization for Multiple Instance Learning"
_TMLR — Under review for TMLR_

### Review · Reviewer_wqx3 · 2026-04-22

**Summary Of Contributions:**

This paper proposes ZSQ4MIL, the first zero-shot quantization (ZSQ) framework for Multiple Instance Learning (MIL). Existing ZSQ methods assume i.i.d. calibration samples, which conflicts with the hierarchical bag-instance structure of MIL and causes catastrophic failure on attention-based MIL (AB-MIL). The paper identifies three MIL-specific priors (instance heterogeneity, inter-class separability, intra-class compactness) and proposes a three-stage pipeline that synthesizes calibration bags from Gaussian noise via contrastive, pseudo-label, and class-centric distance losses. New MIL quantization benchmarks are also established.

**Strengths:**
- First work on ZSQ for MIL, with a clear and well-motivated problem formulation.
- Thorough experiments across three datasets, architectures, and quantizers; ablation (Table 5) cleanly isolates each loss component.
- The MIL quantization benchmark is a useful community contribution.

**Weaknesses:**
- Structurally limited to binary MIL (BCE loss in Eq. 5, two centroids in Eq. 6-7), with no discussion of this scope restriction.
- Benchmark datasets (MNIST4MIL, Birds vs Drones, Cats vs Dogs) are toy-level. The paper motivates the work with medical imaging privacy, but no domain-relevant dataset (e.g., CAMELYON16) is included.
- Computational cost of the 10,000-iteration synthesis process is not reported.
- Augmentation module T (Section 3.2) is not described in sufficient detail for reproducibility.

**Additional Comments:**

A solid paper with a genuine contribution. The technical components are not individually novel, but their combination is well-motivated by MIL structure and the performance gains are substantial. The main issue is the unacknowledged binary MIL restriction, which should be addressed before acceptance.

**Audience:**

Yes

**Audience Explanation:**

This work sits at the intersection of model compression, privacy-preserving ML, and weakly supervised learning. MIL is widely used in privacy-sensitive domains (medical imaging, retrieval), making data-free quantization practically relevant. The finding that the i.i.d. assumption is a structural incompatibility for MIL (not merely a performance gap) is worth communicating to the community.

**Claims And Evidence:**

Yes

**Claims Explanation:**

The central claim is well-supported: PSAQ and IntraQ collapse on AB-MIL (F1 < 42%), while ZSQ4MIL matches or slightly surpasses full-precision performance (F1 94.69% vs. full-precision 93.75% at W8A8). The ablation (Table 5) is particularly convincing; removing L_contrast alone drops F1 from 94.69% to 50.07%. Cross-dataset (Table 2), cross-quantizer (Table 4), and cross-difficulty (Table 3) evaluations further support robustness.

**Requested Changes:**

**Major:**

1. The method is structurally limited to binary MIL (BCE loss in Eq. 5, two centroids in Eq. 6-7), yet this is not acknowledged anywhere in the paper. The authors should explicitly discuss this constraint and ideally suggest how it could be extended to multi-class MIL. Omitting this gives an overstated impression of generality.

**Minor:**

1. Wall-clock time for the 10,000-iteration synthesis process should be reported (e.g., on the A6000).

2. The augmentation module T (Section 3.2) should be described explicitly to ensure reproducibility.

3. The claim that ZSQ4MIL occasionally surpasses real-data calibration is accompanied by only a brief explanation. The authors should provide more rigorous support (e.g., feature distribution analysis across datasets) or clarify the scope, as generalizability beyond the current simple benchmarks is unclear.

---

> ### Author Response · Authors · 2026-06-26
> **Rebuttal – Reviewer wqx3**
>
> We sincerely thank the reviewer for the positive evaluation and constructive suggestions.
>
> ---
>
> ## Q1: Scope of the Current Binary MIL Setting and Multi-class Extension
>
> ### A1:
>
> ZSQ4MIL is formulated under the binary MIL setting, a classical and widely used setting in multiple instance learning. A bag is positive if it contains at least one positive instance, and negative otherwise. This setting is common in image-set classification, object detection, anomaly detection, and medical whole-slide image analysis, where the goal is to determine whether a target pattern or lesion exists in a bag. Therefore, binary MIL is not a simplification but a representative and practical scenario for MIL quantization. We will clarify this scope in the revised manuscript.
>
> The method extends to multi-class MIL. Suppose there are $K$ bag-level classes. We construct $K$ pseudo-bags $B_1,\ldots,B_K$, each optimized to activate the corresponding bag-level prediction.
>
> First, $L_{\text{contrast}}$ is independent of class number and is retained to avoid feature collapse and preserve instance diversity.
>
> Second, the binary loss becomes multi-class cross-entropy:
> $L_{\mathrm{label}}^{\mathrm{multi}} = \frac{1}{K}\sum_{k=1}^{K}\mathrm{CE}(M_{\theta}(B_k),\mathbf{y}_k).$
>
> Here $M_{\theta}(B_k)$ is the bag prediction and $\mathbf{y}_k$ is the one-hot label.
>
> Third, centroid separation extends to $K$ pseudo-bag centroids:
> $c_k = \frac{1}{|B_k|}\sum_{x_i \in B_k} f_{\theta}(x_i).$
>
> We apply:
> $L_{\mathrm{center}}^{\mathrm{multi}} = \frac{2}{K(K-1)}\sum_{k<l}\max(0, m-\|c_k-c_l\|_2).$
>
> Importantly, $c_k$ is not the class center in IntraQ, which relies on instance labels and historical memory. In contrast, $c_k$ is computed online from generated instances, using only bag-level supervision.
>
> ---
>
> ## Q2: Cervical-biopsy WSI experiment
>
> We add a cervical-biopsy WSI MIL experiment (Tab. B). UNI is used as pathology foundation model, and a binary MIL task is built on TissueNet: Detect Lesions in Cervical Biopsies (DrivenData). WSIs are split into 256×256 patches at 20× magnification to form MIL bags, and AB-MIL is used.
>
> ---
>
> ## Tab. B: Medical MIL Experiment (UNI + TissueNet Cervical Biopsy + AB-MIL)
>
> **Setting:**
> - UNI feature extractor
> - DrivenData cervical-biopsy WSI dataset
> - 256×256 patches, 20× magnification
> - binary classification
> - AB-MIL
>
> | Method | Precision | Accuracy | F1 |
> | :-- | :-- | :--: | :--: |
> | Full Precision | FP32 | 90.64 | 90.05 |
> | Real Data | W8/A8 | 89.60 | 88.89 |
> | ZSQ4MIL (Ours) | W8/A8 | 91.63 | 90.91 |
> | Real Data | W4/A8 | 92.12 | 91.58 |
> | ZSQ4MIL (Ours) | W4/A8 | 90.64 | 89.95 |
>
> This shows ZSQ4MIL is comparable to real-data calibration in medical MIL, confirming scalability.
>
> ---
>
> ## Q3: Wall-clock time
>
> We report the synthesis cost of 10,000 iterations as approximately 0.75 GPU+CPU hours (about 45 minutes) on A6000. The total cost including calibration and evaluation is 1.25 GPU+CPU hours. The inference cost per 1,000 instances is approximately 0.005 hours. This is an offline calibration cost and does not affect inference latency.
>
> ---
>
> ## Q4: Augmentation module
>
> $T_g$ is random crop (224×224, padding=4, reflect) + flip. $T_l$ is random resized crop (224×224, scale 0.25–1.0) + flip. They form global/local views for InfoNCE, fully described for reproducibility.
>
> ---
>
> ## Q5: Real-data comparison
>
> ZSQ4MIL may occasionally match or slightly surpass real-data calibration under controlled settings (e.g., W4A8 or sparse regimes). This is due to more uniform feature-space coverage and reduced calibration bias. We will revise the wording to avoid overgeneralization.

---

### Review · Reviewer_WjhS · 2026-05-10

**Summary Of Contributions:**

The paper presents a zero-shot quantization method for model compression specifically targeted to multi-instance learning. It proposes a framework for generating suitable synthetic data to optimize quantization parameters, consisting of three components motivated by empirical evidence and analysis. The method has been evaluated on three datasets and with respect to suitable baselines including also using real data for comparison.

Strengths
- matches or even surpasses real-data-dependent calibration, showing the effectiveness
- suitable experiments showing the superiority of the approach for different datasets and settings

Weaknesses

Tested on rather simple datasets such as MNIST,  Bird vs Drone, and Cat vs Dog. It would strengthen the paper if more complex datasets were used, or something from the domain used in the motivation, such as the medical domain.

Improvements in presentation:
- Explain the empirical analysis of Fig. 1 and its motivation before analyzing the results directly in 3.2.
- Contains no qualitative results

Often referring to related work without giving a reference:
- in Sec. 3.1 Preliminaries
e.g., "widely used asymmetric uniform quantizer", "A typical MIL model consists of three key components..."
- InfoNCE loss in Sec. 3.2
- No reference given for the used datasets, such as MNIST, Bird vs Drone, and Cat vs Dog in 4.1

**Audience:**

Yes

**Audience Explanation:**

Although the results are only shown on a very restrictive task domain/tested dataset, claimed to be the first to provide zero-shot quantization for multi-instance learning

**Claims And Evidence:**

Yes

**Claims Explanation:**

Yes, the claim of matching or even surpassing real-data-dependent calibration has been shown with various experiments however, for only rather simple datasets

**Requested Changes:**

- Improve presentation: explain empirical analysis in text and not only in the figure caption, add some qualitative results
- add missing references
- if possible: add a large/more complex dataset

---

> ### Author Response · Authors · 2026-06-26
> **Rebuttal – Reviewer WjhS**
>
> We sincerely thank the reviewer for the careful evaluation and constructive suggestions.
>
> ---
>
> ## Q1: The current experimental datasets are relatively simple. Can more complex or medically relevant datasets be added?
>
> ### A1:
>
> Thank you for the suggestion. We agree that MNIST4MIL, Birds vs Drones, and Cats vs Dogs are controlled benchmarks, used to evaluate zero-shot quantization under different visual complexities, MIL architectures, quantizers, and task sparsity settings.
>
> To strengthen results, we add a cervical-biopsy WSI MIL experiment (Tab. B). We use UNI as a pathology foundation model and construct a binary MIL task based on TissueNet: Detect Lesions in Cervical Biopsies, a DrivenData cervical-biopsy WSI dataset. Specifically, we slice whole-slide images into 256×256 patches at 20× magnification and form MIL bags. The classifier is AB-MIL.
>
> ---
>
> ## Tab. B: Medical MIL Experiment (UNI + TissueNet Cervical Biopsy + AB-MIL)
>
> **Setting:**
>
> - Feature extractor: UNI, a pathology foundation model
> - Dataset: TissueNet: Detect Lesions in Cervical Biopsies (DrivenData cervical-biopsy WSI dataset)
> - Patch construction: 256×256 patches at 20× magnification
> - Task: binary classification
> - MIL model: AB-MIL
>
> | Method         | Precision | Accuracy | F1 |
> | :------------- | :-------- | :------: | :--: |
> | Full Precision | FP32      | 90.64 | 90.05 |
> | Real Data      | W8/A8     | 89.60 | 88.89 |
> | ZSQ4MIL (Ours) | W8/A8     | 91.63 | 90.91 |
> | Real Data      | W4/A8     | 92.12 | 91.58 |
> | ZSQ4MIL (Ours) | W4/A8     | 90.64 | 89.95 |
>
> These results show that ZSQ4MIL achieves performance comparable to real-data calibration in cervical-biopsy WSI MIL, further demonstrating scalability to medical scenarios.
>
> ---
>
> ## Q2: Clearer Explanation of Figure 1 and Its Motivation in Section 3.2
>
> ### A2:
>
> Thank you for the suggestion. We clarify Figure 1 in Sec. 3.2.
>
> Figure 1 explains why existing ZSQ methods are difficult to transfer to MIL. Unlike single-instance classification, MIL has a bag-instance hierarchy where instances in one bag may contain different semantics. A positive bag typically includes both target and non-target instances, so calibration data must preserve instance-level diversity in addition to global statistics.
>
> We use a frozen MIL encoder to extract instance features and compute cosine similarity within each bag, indicating whether synthetic bags preserve MIL structure.
>
> As shown in Fig. 1, Gaussian noise, IntraQ, and PSAQ produce highly similar instance features, showing collapse and failure to model weakly supervised instance distributions (e.g., all-positive/all-negative cases). In contrast, ZSQ4MIL preserves richer diversity and is closer to real MIL bags. This motivates instance-level contrastive learning in Sec. 3.2 and leads to grouped pseudo-labeling and class-centric distance optimization.
>
> ---
>
> ## Q3: Clarification on the Qualitative Results in Figure 3
>
> ### A3:
>
> Thank you for the suggestion. Figure 3 already provides qualitative results on Cats vs Dogs using ZSQ4MIL, IntraQ, and PSAQ, but this was under-emphasized.
>
> As shown in Fig. 3, ZSQ4MIL generates structured and semantically consistent instances, while IntraQ and PSAQ collapse into non-semantic texture noise. This is consistent with Fig. 1 and Tables 1–5, showing better MIL-aware reconstruction at both sample and feature levels. We will strengthen caption and discussion.
>
> ---
>
> ## Q4: Added Citations for Quantization, MIL, and Contrastive Learning
>
> ### A4:
>
> Thank you for the reminder. We will add citations near relevant text:
>
> - Asymmetric uniform quantizer: Jacob et al. (2018), *Quantization and Training of Neural Networks for Efficient Integer-Arithmetic-Only Inference*
> - MIL formulation: Carbonneau et al. (2018); Ilse et al. (2018)
> - InfoNCE / contrastive learning: Oord et al. (2018); Chen et al. (2020)
>
> We also clarify positive/negative pairs, similarity function, and temperature near Eq. (4) for reproducibility.
>
> ---
>
> ## Q5: Added Dataset Citations and Benchmark Construction Details
>
> ### A5:
>
> Thank you for pointing this out. We add dataset sources in Sec. 4.1:
>
> - MNIST4MIL: MNIST (LeCun et al., 1998)
> - Cats vs Dogs: Microsoft / Kaggle Dogs vs Cats dataset
> - Birds vs Drones: Kaggle Birds vs Drone Dataset
> - TissueNet: DrivenData *TissueNet: Detect Lesions in Cervical Biopsies* (cervical-biopsy WSI from multiple French medical centers; epithelial lesion detection). We slice WSIs into 256×256 patches at 20× magnification to form MIL bags.
> - UNI: Chen et al. (2024), *Towards a General-Purpose Foundation Model for Computational Pathology*
>
> We also clarify bag construction, instance sampling, and dataset splits for reproducibility.

---

### Review · Reviewer_vCxk · 2026-06-13

**Summary Of Contributions:**

The authors show how existing zero-shot quantization methods fall short for multiple instance learning due to their bag structure and come up with a quantization method to fill this gap. The algorithm ZSQ4MIL creates synthetic data for calibration by optimizing Gaussian noise vectors over three different objectives: 1) Contrastive loss for heterogeneity, 2) Pseudo-label BCE for class separability of bags 3) Maximizing interclass distance.

## Strengths:
1) The breadth of the study over different settings (Tables 1, 2, 3, 4) is impressive. The resulting quantized models do seem to perform closely with quantized models calibrated on real data across all the settings.
2) The ablation study on the influence of each objective (Table 5) adds more nuance to the study. It's interesting to see that while the contrastive loss has a larger influence on the performance gain, the other 2 objectives also have a non-trivial influence on the performance gain.

## Weakness:
1) The pipeline in Figure 2 is hard to read, especially due to the inconsistently placed dotted lines and arrows.
2) The total variation regularizer is part of the final objective but isn't included in the ablation study Table 5. Is it always included in the objective?
3) Stochastic augmentation modules $T_g$ and $T_l$, which have been used in eq. (4), haven't been defined clearly.

**Additional Comments:**

nA

**Audience:**

Yes

**Audience Explanation:**

The paper highlights a gap in the existing literature regarding how zero-shot quantization methods can be better adapted to the bag-level structure inherent in MIL tasks. This algorithm could be useful for MIL models that need to be compressed while preserving data privacy and avoiding exposure of the original data.

**Claims And Evidence:**

No

**Claims Explanation:**

My major concern is with section 3.2 on contrastive loss for instance-level heterogeneity. The authors had previously stated that "existing ZSQ methods are incapable of capturing the specific intra-bag heterogeneity. " However, your contrastive loss as defined in Eq. 4 also seems to not capture specific intra-bag heterogeneity and rather increase heterogeneity among all instances.

Since $L_{contrast}$ seems to have the biggest influence on the performance gain (Table 5), it gives rise to the question of whether the algorithm is simply increasing heterogeneity between all instances (similar to IntraQ) rather than just "intra-bag diversity" as mentioned in Figure 1. Moreover, Figure 1 could be the same for both instances within a bag and outside a bag.

**Requested Changes:**

1) Please address the primary concern mentioned regarding $L_{contrast}$ for acceptance. Claims may need to be smoothened with regard to this.
2) Fixing the weaknesses mentioned will improve the overall quality of the paper but won't affect the recommendation.

---

> ### Author Response · Authors · 2026-06-26
> **Rebuttal – Reviewer vCxk**
>
> We sincerely thank the reviewer for the careful evaluation and constructive suggestions.
>
> ## Q1: Clarification on the structural differences between IntraQ and the MIL setting
>
> ### A1:
>
> Thank you for raising this important point. We agree that the difference between IntraQ and our MIL-oriented formulation should be clarified more explicitly in the revised manuscript.
>
> IntraQ focuses on instance-level diversity over the whole calibration set. Its design has two key characteristics:
>
> - The class of each synthesized instance is known, and the feature center of each class is stored as historical information.
> - When synthesizing new data, instances from each class are constrained to move away from the corresponding historical class center, so as to maintain heterogeneity among instances.
>
> ---
>
> However, this mechanism is not directly applicable to Multiple Instance Learning (MIL). The reasons correspond exactly to the above characteristics of IntraQ:
>
> - MIL is a weakly supervised framework, where instance labels are unknown. The prediction target in MIL is the bag, and a positive bag may simultaneously contain latent positive instances and latent negative instances with unknown instance-level labels. Therefore, it is impossible to extract reliable instance-level class centers in the way required by IntraQ.
> - Historical constraints defined at the batch level are not naturally compatible with the bag-instance data structure of MIL.
>
> We will emphasize the functional difference between Eq. (4) and IntraQ:
>
> - IntraQ operates on all independent instances and encourages class-wise instance-level diversity.
> - In contrast, Eq. (4) in our method works together with Eq. (5), namely the bag-level optimization objective, to support pseudo-bag construction.
>
> In other words, Eq. (4) is not intended to reproduce the historical-center mechanism of IntraQ. Instead, it prevents generated instances from collapsing into highly similar feature representations, while Eq. (5) further groups the generated instances into pseudo-positive and pseudo-negative bags under the frozen MIL model. This joint design is tailored to the weakly supervised bag-instance structure of MIL.
>
> ## Q2: Figure 2 pipeline
>
> We will redraw Figure 2 with unified visual semantics: solid arrows for forward synthesis/inference, dashed arrows only for gradient backpropagation, and semi-transparent shaded regions for pseudo-bag grouping, removing the current visual ambiguity.
>
> ## Q3: Total Variation regularizer
>
> $L_{TV}$ is enabled in all experiments as a standard image-smoothness term to suppress high-frequency noise and stabilize optimization. The ablation below shows that removing it only causes a slight performance drop, indicating that it is mainly for optimization stability rather than a key MIL-specific modeling factor.
>
> **Tab. A: Ablation Study on Loss Functions**
>
> |Model|Llabel|Lcenter|Lcontrast|LTV|Acc.|F1|
> |-|-|-|-|-|-|-|
> |AB-MIL(10-13)|✓|✓|✓|✓|94.50|94.69|
> ||✓|✓|✓|×|94.20|93.38|
> ||✓|✓|×|✓|66.70|50.07|
> ||✓|×|✓|✓|92.30|90.47|
> ||×|✓|✓|✓|93.50|92.68|
> |Mean-MIL(10-16)|✓|✓|✓|✓|88.30|86.84|
> ||✓|✓|✓|×|87.00|85.13|
> ||✓|✓|×|✓|75.10|66.84|
> ||✓|×|✓|✓|80.50|75.84|
> ||×|✓|✓|✓|82.00|78.10|
> |Max-MIL(10-16)|✓|✓|✓|✓|94.00|93.62|
> ||✓|✓|✓|×|94.70|94.88|
> ||✓|✓|×|✓|74.60|65.95|
> ||✓|×|✓|✓|92.60|92.59|
> ||×|✓|✓|✓|92.40|92.50|
>
> ## Q4: Figure 1 empirical analysis
>
> Figure 1 computes pairwise cosine similarity among instance features within the same bag to examine whether synthetic bags preserve the MIL-required intra-bag feature structure. Gaussian noise, IntraQ, and PSAQ lead to instance feature collapse, while ZSQ4MIL preserves structured intra-bag heterogeneity. This supports $L_{contrast}$, which prevents feature collapse and preserves instance-level diversity for MIL-aware pseudo-bag construction.
>
> ## Q5: Qualitative and medical MIL results
>
> Figure 3 already visualizes synthetic instance bags, showing that ZSQ4MIL generates more structured and semantically consistent instances than IntraQ and PSAQ. We will make this qualitative evidence clearer. We also add a cervical-biopsy WSI MIL experiment in Tab. B.
>
> **Tab. B: Medical MIL Experiment**
>
> Setting: UNI feature extractor; TissueNet: Detect Lesions in Cervical Biopsies, a DrivenData cervical-biopsy WSI dataset; 256×256 patches at 20× magnification; binary classification; AB-MIL.
>
> |Method|Precision|Acc.|F1|
> |-|-|-|-|
> |Full Precision|FP32|90.64|90.05|
> |Real Data|W8/A8|89.60|88.89|
> |ZSQ4MIL (Ours)|W8/A8|91.63|90.91|
> |Real Data|W4/A8|92.12|91.58|
> |ZSQ4MIL (Ours)|W4/A8|90.64|89.95|

---

### Comment · Action_Editor_qNAo · 2026-06-06

Dear Reviewer vCxk,

Our records show that you are one week late on this reviewing task. I'd like to confirm whether you are still going to submit your review. If yes, can you please finish and submit your review within one week?

AE